# Go Wish Card Game for Meaningful Conversations in the Oncology Healthcare Context: A Narrative Review

**DOI:** 10.3390/cancers17040560

**Published:** 2025-02-07

**Authors:** Bianca Sakamoto Ribeiro Paiva, Fulvio Bergamo Trevizan, Lívia Costa de Oliveira, Karla Santos da Costa Rosa, Vitória Aparecida Betussi, Bruna Minto Lourenço, Miguel Julião, Carlos Eduardo Paiva

**Affiliations:** 1GPQual, Research Group on Palliative Care and Quality of Life, Teaching and Research Institute, Barretos Cancer Hospital, Barretos 14784-400, SP, Brazil; caredupai@gmail.com; 2GPQual, Research Group on Palliative Care and Quality of Life, Barretos Cancer Hospital, Barretos 14784-400, SP, Brazil; fulvio.trevizan@gmail.com (F.B.T.); livia.oliveira@inca.gov.br (L.C.d.O.); karla.rosa@inca.gov.br (K.S.d.C.R.); vitoria.betussi@hotmail.com (V.A.B.); brunaminto65@gmail.com (B.M.L.); 3INCA—National Cancer Institute, Palliative Care Unit, Rio de Janeiro 20560-121, RJ, Brazil; 4Equipa Comunitária de Suporte em Cuidados Paliativos da ULS, Sintra, 2720-276 Amadora, Portugal; miguel.juliao@ulsasi.min-saude.pt

**Keywords:** go wish, games, patient card games, advance care planning, oncology, end-of-life care, share decision-making, autonomy

## Abstract

Many cancer patients face challenging end-of-life care decisions, impacting both themselves and their families. The Go Wish Card Game (GWCG) helps patients communicate their care preferences, improving dialogue with families and healthcare professionals. Studies on GWCG in oncology show that it aids in discussing end-of-life care and enhances patient control. However, barriers such as patient resistance and the need for more training for healthcare professionals exist. Further research, including clinical trials, is needed to fully understand and enhance the effectiveness of GWCG in oncology care.

## 1. Introduction

The incidence and prevalence of cancer, as well as its impact on patients’ lives, highlight the importance of comprehensive care and a broadened perspective for recognizing multiple care demands. Healthcare professionals involved in the care of cancer patients may face challenges related to information, communication, and the encouragement of autonomy. Care directed towards improving patients’ quality of life should be implemented, aiming to provide high-quality care characterized by effective communication, safety, and respect for patient autonomy [1].

Patients diagnosed with cancer are often subjected to aggressive treatments such as radiation therapy, chemotherapy, or surgery, which, despite their benefits, can have various undesirable effects and compromise the patient’s quality of life, potentially leading to treatment abandonment. Therefore, in addition to conventional treatments, it is necessary to explore new care possibilities to minimize the negative impacts of illness and to promote patient autonomy [2].

The development of tools that enable healthcare professionals to identify patients’ wishes throughout treatment and to design interventions that promote their autonomy is a promising strategy for improving the care provided. These tools help to understand the patient’s dignity and can contribute to personalized and humane care by healthcare professionals. They promote active listening, care, the identification and strengthening of patients’ personalities, as well as offering relief, validation, and empowerment [3].

One approach to guiding future patient care involves patients in Advance Care Planning (ACP), where they can create an Advance Directive (AD) based on their desires and core values. The AD facilitates more personalized care by sharing patient decisions with the healthcare team and respecting their autonomy through an ongoing, dynamic process that can be reviewed and modified as the disease progresses and treatment evolves, whether curative or palliative [4]. Another useful tool for identifying patients’ primary desires is the Go Wish Card Game (GWCG) [5].

There are other games to guide conversations for ACP, such as Hello [6,7], Hopewell Hospitalist [8], the Life unlocking card game [9], and the Pallium game [10]. However, these games are not available in many languages and have not been evaluated in the oncology setting, as GWCG has been adapted for ten different cultures [5] and with research in oncology [11].

In our search for scientific evidence on the GWCG, we discovered a notable gap in the literature. Although we reviewed a range of sources, including qualitative studies and reviews, only a small fraction of the research focused on intervention outcomes. This highlights a significant lack of comprehensive studies specifically evaluating the GWCG’s effectiveness, particularly in end-of-life care settings [11,12,13,14,15,16,17,18,19,20,21,22,23,24,25,26,27,28,29,30,31,32,33,34,35,36,37,38,39,40,41,42,43,44,45,46,47,48,49,50,51] (Appendix A). SANRA guidelines were used to ensure the quality of reporting [52].

The limited research available underscores the urgent need for clear guidance from professionals on the use and effectiveness of the GWCG to ensure it is applied effectively and practically in various care environments.

This article will methodically examine the implications of using GWCG in clinical oncology practice. We will discuss the origins, objectives, relevance, challenges, and evidence supporting the tool. Additionally, we will develop a conceptual model to guide healthcare professionals in the application of GWCG.

## 2. What Is the Go Wish Card Game and Who Is It Intended for?

GWCG was developed in the United States in the 1990s by Coda Alliance, a nonprofit organization [22]. Since then, the tool has been culturally adapted, translated, and used in various countries (Coda Alliance) to explore deep and subjective themes related to the preferences and wishes of patients with chronic or advanced illnesses at the end-of-life (EoL) stage [23]. In some countries, there are partners authorized to sell the game, including a guide for its use [5]. GWCG is simple, easy to use, and requires minimal training to promote and facilitate discussions about patient’s values and goals [44]. It facilitates the exchange of thoughts and feelings among patients, ensuring that their preferences are respected [44].

The game consists of 36 cards and includes a guidance manual. The cards propose a patient-centered approach by suggesting important goals and values, special needs, care expectations, desires, and the promotion of shared decision-making [22]. One card serves as a wildcard, allowing the patient to express something not covered by the other cards, while the remaining 35 cards describe the common wishes of individuals who are seriously ill or at EoL. Patients categorize each card into one of three levels of importance: “very important”, “somewhat important”, or “not important”. The 10 cards classified as “very important” are then prioritized and discussed with the healthcare professional [23]. These results serve as a basis for reflection, discussion, and engagement in end-of-life care planning and the promotion of dignity [44].

The Chinese version of the GWCG, known as the Heart to Heart Card Game (HHCG), is structured differently but maintains the premise of a personalized approach to care. HHCG consists of 54 cards, organized into four suits representing needs from different domains: 13 Spades (physical needs), 13 Hearts (spiritual needs), 13 Diamonds (financial needs), 13 Clubs (social needs), and two Jokers (Special Wishes cards). Each category of needs includes 13 different questions. The two “Special Wishes” cards are blank, allowing participants to add personal questions not covered by the other cards. The gameplay involves selecting 12 important cards, three from each suit. From these 12 cards, participants are instructed to choose the three that are most important to them [51].

## 3. What Is the Relevance of the Go Wish Card Game in Clinical Oncology?

The relevance of using GWCG in clinical practice lies in its ability to address the resistance often observed among patients and families when discussing end-of-life issues [11]. The active engagement through the wishes expressed in the GWCG effectively prompts meaningful discussions with cancer patients in end-of-life contexts, who are frequently fatigued by their illness and treatment [23,44,51].

Conversations about EOL priorities are considered beneficial for patients with advanced cancer [30]. In this context, the use of the GWCG can help them articulate thoughts, feelings, and memories, as well as aid in understanding and deciding what is most important to them [11]. In a study involving 346 cancer patients, 97.9% assigned high priority to the card about the desire to participate in ACP, and 95.7% assigned the same level of priority to the card about the desire for their family to respect their wishes. In addition to assisting with ACP, the GWCG facilitated the creation of ADs and alleviated discomfort when talking and thinking about death [25].

As an interactive communication method, the use of GWCG promotes opportunities for patients to reflect and speak, facilitating the expression of their wishes and preferences regarding EoL [11]. They are especially beneficial for hospitalized patients, undergoing prolonged infusion treatments or experiencing long waiting periods between consultations. Patients’ time is precious, and any intervention must carefully maximize usefulness and enjoyment [45].

The scientific literature highlights that the card game has been crucial in some care situations. One example is the case of Ms. M, a 69-year-old with acute myeloid leukemia, whose family was reluctant to discuss end-of-life issues. The healthcare team introduced the GWCG, making it available for use whenever the patient wished to address the topic. Her children reported that the straightforward statements in the cards helped them discuss hopes and wishes with the patient, aligning decisions about comfort measures [44].

Additionally, patient preferences may vary culturally. For Chinese patients, family involvement is crucial in end-of-life care, unlike the predominantly white American population, which does not prioritize this aspect for a good death [26]. American patients with advanced cancer often place higher importance on religious aspects. These differences reflect cultural values: Chinese culture emphasizes family and harmony, while Western culture emphasizes individualism and autonomy [26].

In terms of accessibility, games can transform the complexity of managing serious illnesses into actionable steps, providing patients with a representational framework of their experience while also being geared towards achieving their health-related goals [23]. Additionally, games offer distraction, education, entertainment, and even social connection [44].

An evaluation of the Chinese version of the GWCG (HHCG) revealed that, among 40 participants, 38 (95%) considered it relevant to discuss end-of-life issues and felt that their opinions were respected. Additionally, 35 participants (87.5%) found that the game facilitated discussions about these issues and helped them express their preferences. Furthermore, 35 participants (87.5%) indicated that they would recommend the HHCG to others, and 29 (72.5%) expressed a willingness to engage in discussions about ADs. More than 70% of participants rated the HHCG positively, suggesting that it may be used as a communication tool to encourage end-of-life discussions between cancer patients and healthcare professionals [26].

The HHCG emphasizes the importance of identifying patient preferences regarding EoL care and values relevant to Chinese culture, while also encouraging a more holistic view of the patient by addressing a range of physical, spiritual, social, and financial needs. Additionally, since no patient selected the special wish cards, it suggests that the other cards were sufficient to address the end-of-life needs of the participants [26].

To illustrate this study’s findings, a conceptual model (Figure 1) was developed, emphasizing the use of the GWCG in relation to cancer, treatment, and overall patient well-being. The model highlights expected outcomes in both quality of life and quality of death through the use of this visual, interactive, and engaging tool. It demonstrates that a patient-centered approach, incorporating the GWCG, is essential for addressing patient preferences, improving communication, and ensuring more personalized care, particularly in end-of-life decision-making. By integrating the GWCG into the treatment process, the model underscores the importance of managing and mitigating negative outcomes, such as treatment abandonment and decision-making difficulties.

## 4. How and When Should the Professional Introduce the Go Wish Card Game to the Patient?

As an innovative tool in oncological practice, the GWCG may be highly beneficial in contexts where patient autonomy and dignity are prioritized [22]. This strategy can be applied at any stage of treatment, whether as the EoL process approaches or simply to share the wishes and desires of cancer patients [23]. It facilitates shared decision-making with the healthcare team and loved ones, provided that all parties demonstrate readiness for such discussions [31,44].

Thus, its use should be carefully planned, taking into account the appropriate timing for introduction, the patient’s emotional state, and the stage of oncological treatment. In some cases, this moment should follow an initial emotional stabilization after diagnosis or may be more suitable during transitions in treatment, such as when a change in therapeutic strategy is needed [11]. It is important that the introduction of the GWCG occurs in a safe and comfortable environment where the patient feels at ease to express their feelings and concerns [53].

Starting the game with a patient requires sensitivity and specific skills from the professional, regardless of their field of expertise [54]. The objective of the game should be explained clearly and simply, emphasizing that its purpose is to help identify and prioritize the patient’s wishes regarding their care and EoL decisions [54]. It is crucial for the professional to remain attentive to the patient’s emotional state throughout the process, even after the game has commenced. This vigilance serves a protective role, preventing the continuation of the game during periods of emotional instability, high anxiety, or fear, or when the patient is dealing with newly received and potentially stressful news. Managing the game in an oncological context requires understanding and consideration of the emotional and clinical instabilities of cancer patients. Thus, the approach should be gradual, allowing the patient to control the pace of the interaction [55].

Among the specific skills expected of professionals using the GWCG are empathy, compassion, active listening, and effective communication, which are essential for creating and maintaining a trusting environment [56]. Additionally, these professionals must be adept at handling sensitive topics, such as death and dying, in a respectful and non-judgmental manner, providing a space where the patient feels validated in their emotions and choices [4]. The ability to recognize and respect others’ boundaries is crucial to ensuring that the use of the GWCG remains a therapeutic and non-intrusive experience [11].

Regarding patients, it is possible to identify those who may be more receptive to discussing their wishes using the GWCG [11]. Patients who are more likely to engage are generally those who show an active interest in participating in decisions about their care, have undergone a process of acceptance of their condition, and have adequate emotional support [51]. Patients in a state of denial, with high levels of anxiety, or in the early stages of accepting their diagnosis may resist participation, necessitating a more cautious approach and, often, postponement of the use of the tool until they are better emotionally prepared [57].

The sensitivity and accuracy of healthcare professionals in recommending the use of the GWCG for cancer patients can ensure that the intervention is more appropriate and respectful [57]. These aspects help in recognizing signs of readiness for such discussions, thereby avoiding premature conversations that could induce unnecessary anxiety [58]. An untimely approach, without attention to the aforementioned details, can trigger fears and insecurities, while a delayed approach might leave the patient without the opportunity to express their preferences at a crucial time [12]. Therefore, the professional should be attuned to the patient’s timing, carefully observing signs of acceptance, openness to dialogue, and comfort levels when discussing topics related to death [57].

In summary, respecting the patient’s wishes should be a priority throughout the entire process, from eligibility for the game to its conclusion. The professional should value and support the patient’s decisions, even if the patient chooses to decline discussing the matter. Forcing participation or insisting on an inappropriate moment can violate the respect for patient autonomy and compromise the relationship with the healthcare team. The professional must be prepared to accept that some patients, especially those in very advanced stages or actively dying, may prefer not to engage in these discussions. In such cases, it is crucial for the healthcare team to provide emotional support, respect the patient’s silence, and seek other ways to ensure that the care provided remains centered on the patient’s wishes and needs, even if not expressed through the GWCG (Figure 2).

## 5. What Are the Challenges and Barriers to Implementing the Go Wish Card Game in Oncology Clinical Practice?

The implementation of GWCG in oncology practice encounters challenges that must be addressed effectively [11]. Discussions regarding EoL care necessitate a sensitive approach and a significant investment of time and emotional support from healthcare professionals and patients [22]. Additionally, there are specific challenges and barriers associated with the use of GWCG in clinical practice [11,22].

Regarding cancer patients, there may be discomfort during the conversation, exacerbated by a lack of clarity about their priorities or emotional fragility in engaging in the discussion [54]. The patient’s perception of their prognosis and the cultural stigma surrounding discussions about death can present significant obstacles, manifesting as resistance and denial [22]. Additionally, a lack of familiarity with the tool or the perception that it is intrusive may lead to refusal of the intervention. These challenges can be minimized through effective collaboration in building relationships between patients and healthcare professionals [27].

An important aspect that may become a significant barrier is the central role of the relationship in the application of the GWCG. A study demonstrated that patient adherence to the tool was considerably higher when the GWCG was administered by a professional with whom the patient already had an established relationship. This suggests that the success of the intervention may be closely related to the level of trust and the quality of the existing therapeutic relationship [28].

Overcoming the barrier of the relationship between the healthcare professional and the patient involves more than just communicating information; it encompasses empathy, mutual understanding, and a sense of security that enables the patient to comfortably explore sensitive issues [11,26,52]. When administered by a trusted professional, the patient is more likely to engage openly and sincerely, believing that their concerns will be addressed with respect and sensitivity [45,58].

Discussions facilitated by the GWCG tend to evolve over multiple consultations, making it crucial to consider not only the emotional aspects but also the patient’s clinical conditions. The tool can become challenging to apply when the patient is debilitated, with compromised disposition and cognition. This highlights the need to introduce EoL conversations early, from the outset of the illness. Patients with more debilitated clinical conditions may report fatigue when selecting the cards, difficulties handling them, remembering the written content, and recalling how they were organizing them [19].

Regarding barriers, the patient’s perception of their prognosis and the stigma associated with death can limit their engagement with the GWCG due to cultural, personal, and family factors. Additionally, the difficulty that healthcare professionals often face in dealing with EoL issues can impact discussions about EoL wishes [11]. Guidance from qualified professionals in using the game can significantly enhance its applicability and the formulation of AD [25].

On the other hand, healthcare professionals and teams face significant barriers to using the GWCG. A lack of training in communication and empathy, which are crucial for discussing sensitive topics such as EoL care, is a major issue [4]. Additionally, natural resistance from some professionals to dealing with EoL issues and the workload burden can lead to reluctance in introducing the GWCG [11]. The lack of time and adequate support from healthcare institutions limits the integration of the tool into daily practice, resulting in ineffective or nonexistent implementation in most oncology centers [55,56,57].

In addition to the mentioned barriers, family-related obstacles play a critical role in the implementation of the GWCG. Many families struggle to accept the patient’s condition and may discourage discussions about EoL care, viewing them as unnecessary or even detrimental to emotional well-being and denying the reality of death. Culturally, the taboo surrounding death leads to a “conspiracy of silence”, which reduces receptivity to the game [11]. In such moments, there is a breakdown in communication between the family and the patient, as an atmosphere of secrecy is established, especially regarding the patient’s actual clinical condition. Instead of security and serenity, silence creates a heightened state of anxiety, fear, and confusion. Furthermore, it denies individuals the opportunity to reframe and plan their lives based on plausible hopes. It is even more difficult for health professionals to discuss end-of-life care when there is no support from the family or the patients themselves who are unable to address the subject. It is essential that the team guides the family on the importance of this discussion and acts as an intermediary, interacting with the family members so that they can also express their wishes [59].

Institutional barriers also come into play, as the lack of clear policies and organizational support limits the adoption of the GWCG, making it a seldom-practiced tool [22,23]. Institutional resistance is further exacerbated by the lack of robust scientific evidence on the benefits of the GWCG, which hinders its acceptance and widespread use as an effective tool in oncology care [31,44].

In an institutional context, integrating the GWCG into existing care plans is challenging, particularly in healthcare environments with established and dynamic routines. The main obstacle is incorporating the tool without causing significant disruptions or adding extra burdens for professionals and patients. For the GWCG to be effective, it must complement the existing care without substantially altering daily practices. This requires effort to adapt the tool to the needs and workflows of the clinical setting, ensuring it is seen as a natural extension of patient care. The game should be strategically implemented during appropriate times, such as routine consultations or follow-up meetings, so as not to interfere with other important procedures.

Therefore, to overcome the discussed challenges and barriers, it is essential to develop clinical and communication skills, awareness of prognosis, and cultural sensitivity, and to create a supportive care environment that promotes the patient’s overall well-being. In this way, the use of the GWCG becomes a natural extension of care, reinforcing the role of the therapeutic relationship in facilitating meaningful conversations and supporting informed patient decision-making. Table 1 summarizes the challenges and barriers to implementing the GWCG in clinical practice, along with strategies for overcoming them.

## 6. Conclusions

The GWCG is demonstrated to be an effective tool for facilitating end-of-life discussions in oncology, promoting open communication between patients and their families. However, its implementation faces significant barriers, such as patient resistance, lack of knowledge and training among professionals, and inadequate institutional support. Additionally, it is crucial to gradually introduce EoL discussions, building trust over time and respecting the patient’s emotional state while also promoting cultural sensitivity and framing the GWCG as a tool for exploring personal values rather than solely focusing on death. Studies with higher levels of scientific evidence, such as clinical trials, are needed to validate and expand the tool’s applicability. Overcoming these barriers and conducting robust research can ensure that the GWCG is effectively integrated into oncology care, facilitating shared decision-making about EoL issues.

## Figures and Tables

**Figure 1 cancers-17-00560-f001:**
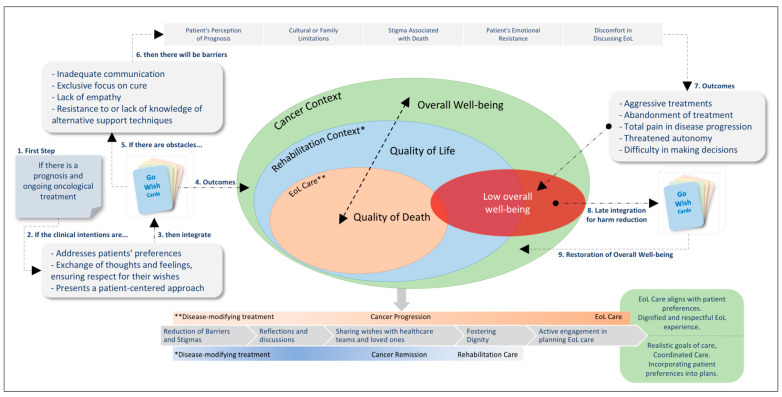
Integrated conceptual model for the use of Go Wish Card Game: oncological scenarios and application strategies in rehabilitation and end-of-life care. Note: EoL Care: End-of-Life Care.

**Figure 2 cancers-17-00560-f002:**
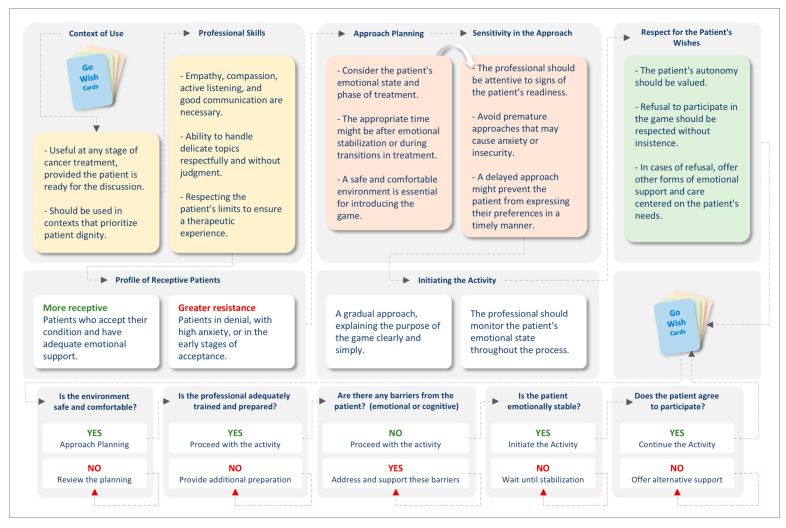
When and how to implement the Go Wish Card Game in oncological care: a clinical decision-making flowchart. Note: GWCG: Go Wish Card Game.

**Table 1 cancers-17-00560-t001:** Challenges and barriers to implementing the Go Wish Card Game in clinical practice and strategies for overcoming them.

In Which Context?	What Is the Barrier?	What Is the Resolution Strategy?
**Patient Barriers**	**Discomfort with EoL Discussions** Patients may feel emotionally fragile or lack clarity about their EoL priorities, leading to discomfort when discussing these topics.	Gradually introduce EoL discussions, building trust over time and ensuring the patient’s emotional state is respected. Provide emotional support and reassurance that their concerns are valid and valued.
	**Perception of Prognosis and Cultural Stigma** Cultural factors and personal denial about death can make patients resistant to the GWCG.	Enhance cultural sensitivity in conversations and frame the GWCG as a tool for exploring personal values rather than focusing solely on death. Offer multiple opportunities for discussion to reduce resistance.
	**Debilitated Clinical Condition** Physical and cognitive limitations may prevent patients from fully engaging with the GWCG.	Introduce the tool early in the disease process when patients are more likely to be physically and cognitively capable of participation. Adapt the tool’s format for patients experiencing fatigue or cognitive challenges.
**Health Professional Barriers**	**Lack of Training in Communication and Empathy** Many healthcare professionals are not adequately trained to discuss sensitive EoL topics, leading to avoidance or reluctance to use the GWCG.	Implement targeted training programs focusing on communication skills, empathy, and handling EoL conversations. Encourage role-playing scenarios to help professionals become comfortable with these discussions.
	**Workload and Time Constraints** High patient loads and limited time hinder healthcare professionals from dedicating the necessary time to introduce and engage in discussions using the GWCG.	Integrate the GWCG into routine care, allowing for gradual introduction during follow-up consultations. Provide institutional support to allocate specific times for these conversations, such as during less intensive clinical visits.
	**Resistance to Discussing EoL Issues** Some professionals may naturally resist discussing EoL matters, viewing them as uncomfortable or unnecessary.	Normalize EoL discussions within the clinical setting by highlighting the positive impact on patient care. Promote the GWCG as a standard part of care, emphasizing its value in helping patients express their wishes.
**Family Barriers**	**Family Resistance to EoL Conversations** Families may oppose EoL discussions, believing that they may cause emotional harm or hasten a negative outlook on the patient’s condition.	Educate families on the importance of EoL planning for enhancing patient autonomy and well-being. Involve them in the conversation early on, addressing their concerns in a compassionate manner.
	**Cultural and Emotional Taboo Around Death** Some families may adhere to cultural beliefs that avoid or stigmatize conversations about death, creating a “conspiracy of silence”.	Approach the topic with cultural sensitivity and explain that the GWCG is a tool for discussing personal values and wishes, not just death. Encourage family participation in these conversations to foster openness.
**Institutional Barriers**	**Lack of Organizational Support** Many healthcare institutions do not have clear policies or support for integrating the GWCG into patient care, leading to sporadic or nonexistent use.	Advocate for the development of institutional policies that incorporate the GWCG as part of standard oncology care. Demonstrate the benefits through evidence-based research to gain institutional buy-in.
	**Disruption to Routine Care Practices** The introduction of the GWCG may be seen as a disruption to existing clinical workflows, making professionals hesitant to adopt it.	Position the GWCG as an extension of patient care rather than an additional burden. Incorporate it into already scheduled patient interactions, ensuring that it complements existing routines without adding significant time demands.
	**Lack of Robust Scientific Evidence** Limited evidence supporting the effectiveness of the GWCG may hinder its widespread adoption.	Encourage research that demonstrates the benefits of the GWCG in oncology care, focusing on its role in improving patient outcomes and decision-making processes. Present findings to institutional leaders to promote its integration.

Note: the guidelines described in Table 1 are derived from the main scientific evidence found in this narrative review, in order to highlight important strategies for implementing the GWCG in clinical practice, despite its challenges and barriers. GWCG: Go Wish Card Game.

## Data Availability

Not applicable.

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
