# Peer review of "Go Wish Card Game for Meaningful Conversations in the Oncology Healthcare Context: A Narrative Review"

_cancers, 2025, doi:10.3390/cancers17040560_

Round 1
Reviewer 1 Report
Comments and Suggestions for Authors
This is a well written and well structured review of the advantages and challenges of the Go Wish Cards Game. Eventhough I never heard about this game (I think it is not or hardly being used in my country, the Netherlands), as a researcher in meaningful communication, advance care planning and shared decision making I recognise all the advantages and challenges, as they count in general when it concerns contextualised communication and care.
I have some suggestions:
In the introduction, the GWCG is introduced, but it takes half a page before it is explained what it is. Please adapt.
Figure 1 on page 3/12 is very complex. I really don't know where to start.
It does not become clear from the review how this game relates to other methods to explore patient preferences. If there are no studies that provide such insights, please mention this in the discussion.
In my research experience, many patients don't explicitly realise which of their personal context is important to share with their healthcare professional. They do share a lot of 'red flags' that might refer to essential patient context, but if the healthcare professional does not explore thes red flags, the possible essential context remains hidden. (See papers of Saul Weiner and Alan Schartz). By showing cards that not directly refer to specific meaningful aspects in the life of a patient, it might be too difficult for a patient to relate the card to their specific context. is there any literature on this game play and the 'translation' from a generic topic to specific essential patient context? The fact that no patient selected the special wish card, might also mean that they don't realise what their special needs are. (van Meurs et al. Pall Med 2022 36(10):1493-1503. Van Meurs et al. Cancer Nursing 2018 41(4):E39-E45.
On some places in the text, it is spelled 'advanced care planning' which should be 'advance care planning' (also in the key word)
Author Response
REVIEWER 1
This is a well written and well structured review of the advantages and challenges of the Go Wish Cards Game. Eventhough I never heard about this game (I think it is not or hardly being used in my country, the Netherlands), as a researcher in meaningful communication, advance care planning and shared decision making I recognise all the advantages and challenges, as they count in general when it concerns contextualised communication and care.
I have some suggestions:
- In the introduction, the GWCG is introduced, but it takes half a page before it is explained what it is. Please adapt.
Response: Thank you for your suggestion. The explanation of what the Go Wish Card Game is can be found on page 3, item 2 of the manuscript.
- Figure 1 on page 3/12 is very complex. I really don't know where to start.
Response: Thank you for your valuable feedback regarding Figure 1. We carefully reviewed the figure and made several adjustments to improve its clarity. We added sequential numbering to guide the step-by-step process, reduced the number of arrows and relationships to simplify the visual layout, and included additional legends for better understanding. Additionally, as suggested by one of the reviewers, we described the barriers in separate boxes to clarify that they are not sequential, addressing the valid concern that the previous layout might have suggested otherwise. Furthermore, as suggested by two reviewers, we repositioned the figure later in the text to improve the contextual flow. We hope these revisions address your concerns and enhance the overall presentation.
- It does not become clear from the review how this game relates to other methods to explore patient preferences. If there are no studies that provide such insights, please mention this in the discussion.
Response: GWCG is a simple method, easy to use, require minimal training to promote and facilitate discussions about patient’s values and goals (Menkin, 2007) [43]. (page 3, item 2)
- In my research experience, many patients don't explicitly realise which of their personal context is important to share with their healthcare professional. They do share a lot of 'red flags' that might refer to essential patient context, but if the healthcare professional does not explore thes red flags, the possible essential context remains hidden. (See papers of Saul Weiner and Alan Schartz). By showing cards that not directly refer to specific meaningful aspects in the life of a patient, it might be too difficult for a patient to relate the card to their specific context. is there any literature on this game play and the 'translation' from a generic topic to specific essential patient context? The fact that no patient selected the special wish card, might also mean that they don't realise what their special needs are. (van Meurs et al. Pall Med 2022 36(10):1493-1503. Van Meurs et al. Cancer Nursing 2018 41(4):E39-E45.
Response: Thank you for your observation. Indeed, the lack of attention to the patient's context, as suggested by Saul Weiner and Alan Schwartz, is often an underestimated cause, known as "contextual error." Detecting this error usually requires careful listening during the consultation to perceive important aspects of the patient's context that may affect treatment adherence and health outcomes. In the case of the Go Wish Card Game, the difficulty some patients may have in relating the cards to their own specific context could be seen as a manifestation of this contextual error, if the healthcare professional does not adequately explore the patient's contextual factors.
Additionally, as you mentioned, the absence of selecting the special wish card may reflect a lack of awareness on the patient's part regarding their own needs. This aligns with evidence suggesting that attention to the patient's context can improve health outcomes and reduce errors in care. Although the literature on the "translation" of generic themes into the specific context of each patient is still in its early stages, research suggests that strategies like the GWCG can help raise awareness for both patients and professionals on these issues.
- On some places in the text, it is spelled 'advanced care planning' which should be 'advance care planning' (also in the key word)
Response: Thank you for your suggestion. We have made adjustments in the text and in the keywords.
We sincerely appreciate all the suggestions and feedback, which were crucial in helping us improve the article

Reviewer 2 Report
Comments and Suggestions for Authors
This is an interesting article that describes how to use the GoWishCardsGame.
It is well organised with appropriate section headings, well written and easy to follow. It is written in a manner that affords understanding.
Essentially it is a review that outlines how the game should be introduced to patients when end of life could be discussed. The article describes the relevance of the game in Oncology and the barriers associated with its use. It outlines methods that might overcome these barriers. My main concern is that much of what is written could be applied to any method for discussing end of life care. There isn't a lot of new information for the Oncology team who will be familiar with approaching discussions about end of life.
In the Introduction the authors state that they will examine the implications of using the game in clinical Oncology practice. Unfortunately I could find little evidence of this that was specific to the game. it would have been very useful and interesting for the authors to have a section that discussed what unique characteristics or advantages the game brings to the discussion (in comparison to other methods for discussing end of life care). There are a lot of useful references that might contain this information, yet this isn't brought out distinctly within the article. In this regard, Fig 1 attempts to place the game in context, but still doesn't directly address this area. Incidentally, I would place Fig 1 much later in the document, as a summary, rather than at the beginning of the article.
I hope these comments are useful to the authors who have clearly put a lot of work into this review.
Author Response
REVIEWER 2
This is an interesting article that describes how to use the Go Wish Cards Game.
It is well organised with appropriate section headings, well written and easy to follow. It is written in a manner that affords understanding.
- Essentially it is a review that outlines how the game should be introduced to patients when end of life could be discussed. The article describes the relevance of the game in Oncology and the barriers associated with its use. It outlines methods that might overcome these barriers. My main concern is that much of what is written could be applied to any method for discussing end of life care. There isn't a lot of new information for the Oncology team who will be familiar with approaching discussions about end of life.
Response: Thank you for your concern. However, the article specifically addresses all aspects related to the Go Wish Card Game, including how it can guide the healthcare team, especially those who face significant difficulties when it comes to discussing end-of-life issues. We understand your point, but the article provides a very detailed context to help healthcare professionals who, although they may be aware of these issues, often struggle to put them into practice. Our goal is to provide clear guidance for these professionals to help them navigate these difficult conversations.
- In the Introduction the authors state that they will examine the implications of using the game in clinical Oncology practice. Unfortunately I could find little evidence of this that was specific to the game. it would have been very useful and interesting for the authors to have a section that discussed what unique characteristics or advantages the game brings to the discussion (in comparison to other methods for discussing end of life care). There are a lot of useful references that might contain this information, yet this isn't brought out distinctly within the article.
Response: Thank you for your suggestion. We have included this paragraph in the introduction, on page 2.
There are other games to guide conversations for ACP, such as Hello [6,7], Hopewell Hospitalist [8], Life unlocking card game [9] and Pallium Game [10]. However, these games are not available in many languages and have not been evaluated in the oncology setting, as GWCG has been adapted for ten different cultures [5] and with research in oncology [11].
We added these references:
[6] Van Scoy, L.J.; Watson-Martin, E.; Bohr, T.A.; Levi, B.H.; Green, M.J. End-of-Life Conversation Game Increases Confidence for Having End-of-Life Conversations for Chaplains-in-Training. Am J Hosp Palliat Care. 2018, 35, 592-600. c10.1177/1049909117723619.
[7] Van Scoy, L.J.; Levi, B.H.; Bramble, C.; Calo, W.; Chinchilli, V.M.; Currin, L.; Grant, D.; Hollenbeak, C.; Katsaros, M.; Marlin, S.; Scott, A.M.; Tucci, A.; VanDyke, E.; Wasserman, E.; Witt, P.; Green, M.J. Comparing two advance care planning conversation activities to motivate advance directive completion in underserved communities across the USA: The Project Talk Trial study protocol for a cluster, randomized controlled trial. Trials. 2022, 23, 829. https://doi.org/10.1186/s13063-022-06746-3.
[8] Mohan, D.; O'Malley, A.J.; Chelen, J.; MacMartin, M.; Murphy, M.; Rudolph, M.; Barnato, A. Videogame intervention to increase advance care planning conversations by hospitalists with older adults: study protocol for a stepped-wedge clinical trial. BMJ Open. 2021, 11, e045084. https://doi.org/10.1136/bmjopen-2020-045084.
[9] Phenwan, T.; Apichanakulchai,T.; Sittiwantana, E. Life unlocking card game in death and dying classroom for medical students. MedEdPublish (2016). 2018, 7, 181. https://doi.org/10.15694/mep.2018.0000181.1.
[10] Fernandes, C.S.; Vale, M.B.; Magalhães, B.; Castro, J.P.; Azevedo, M.D.; Lourenço, M. Developing a Card Game for Assessment and Intervention in the Person and the Family in Palliative Care: "Pallium Game". Int J Environ Res Public Health. 2023, 20, 1449. https://doi.org/10.3390/ijerph20021449.
- In this regard, Fig 1 attempts to place the game in context, but still doesn't directly address this area. Incidentally, I would place Fig 1 much later in the document, as a summary, rather than at the beginning of the article.
Response: Thank you for your thoughtful suggestion. We greatly appreciate your detailed feedback on Figure 1. We recognize the importance of positioning the game more clearly within its context and have made revisions to the figure to better address this. Specifically, we have added legends and supplementary labels to clarify the connection between the game and the broader context of the study, ensuring its relevance is more apparent. Additionally, in response to your suggestion, we have moved Figure 1 later in the manuscript so that it functions as a summary, rather than appearing at the start of the article. We agree that this change enhances the logical flow of the manuscript and results in a more cohesive structure.
I hope these comments are useful to the authors who have clearly put a lot of work into this review.
Response: We sincerely appreciate all the suggestions and feedback, which were crucial in helping us improve the article
Reviewer 3 Report
Comments and Suggestions for Authors
This is a well-written narrative review, with nicely designed (and informative) figures and table. I enjoyed reading it. I do not have any major concerns, but I have suggested some changes which I believe can strengthen the manuscript.
1. While the assessment of narrative reviews is not as stringent as the assessment of systematic reviews, I would suggest taking a quick look at the SANRA checklist (https://pmc.ncbi.nlm.nih.gov/articles/PMC6434870/pdf/41073_2019_Article_64.pdf). It would not take much work to ensure that all items score well – items 3 (description of the literature search) and 4 (referencing) are the ones that need most attention. Then, you can mention in the manuscript that SANRA was used to ensure quality of reporting.
1a. The manuscript reads well as it is, so perhaps an additional description of the literature search can be added to the supplementary file?
1b. Regarding referencing; my suggestion is to provide in-text citations more consistently/accurately. Several paragraphs have a group of references at the end, and it is not clear where evidence presented in different sentences came from. For example, where did the term “conspiracy of silence” (page 7, line 291) come from? Because all references are at the end of the paragraph, the source is not clear.
2. Placement of Figure 1 and grey boxes at the top:
2a. I wondered whether Figure 1 should be presented a bit later, as it makes much more sense after explanations about the Go Wish Cards Game (GWCG). The last sentence in the introduction could remain the same, but without “(Figure 1)”. The figure could be added a bit later, perhaps with the addition of two or three sentences summarising its key points?
2b. Would the grey items on top look better in boxes? The way they are displayed, it looks like there is a specific order/sequence to them. From what I understood this is not the case and this is just a list (in no particular order)?
3. Page 3 of 12, line 104: it may be best to add a subject to the sentence, either repeating GWCG, or if wishing to avoid repetition, “The cards game consists of…”
4. Page 7 of 12, line 276: Perhaps “family factors” instead of “familial factors”? The latter seems to indicate this is about diseases or disorders that run in families.
5. Table 1, page 8 of 12: I like this table, but it is not clear if it is derived solely from the evidence, if the resolution strategies have been developed by the manuscript authors, or maybe a combination of both. It would be useful to clarify this, even if as a footnote.
6. Overall: Are the contents from the cards copyrighted? A text box with the items would be very informative, although I understand this may not be possible.
Author Response
REVIEWER 3
This is a well-written narrative review, with nicely designed (and informative) figures and table. I enjoyed reading it. I do not have any major concerns, but I have suggested some changes which I believe can strengthen the manuscript.
- While the assessment of narrative reviews is not as stringent as the assessment of systematic reviews, I would suggest taking a quick look at the SANRA checklist (https://pmc.ncbi.nlm.nih.gov/articles/PMC6434870/pdf/41073_2019_Article_64.pdf). It would not take much work to ensure that all items score well – items 3 (description of the literature search) and 4 (referencing) are the ones that need most attention. Then, you can mention in the manuscript that SANRA was used to ensure quality of reporting.
Response: We thank the reviewer for their comment. We have reviewed the SANRA guidelines and have referenced them in the introduction (page 2).
1a. The manuscript reads well as it is, so perhaps an additional description of the literature search can be added to the supplementary file?
Response: We add a description of the literature search in the supplementary material 1.
1b. Regarding referencing; my suggestion is to provide in-text citations more consistently/accurately. Several paragraphs have a group of references at the end, and it is not clear where evidence presented in different sentences came from. For example, where did the term “conspiracy of silence” (page 7, line 291) come from? Because all references are at the end of the paragraph, the source is not clear.
Response: The prompts have been restructured as suggested. A reference has been inserted to better describe the term “conspiracy of silence” (page 8): In such moments, there is a breakdown in communication between the family and the patient, as an atmosphere of secrecy is established, especially regarding the patient’s actual clinical condition. Instead of security and serenity, silence creates a heightened state of anxiety, fear, and confusion. Furthermore, it denies individuals the opportunity to reframe and plan their lives based on plausible hopes. It is even more difficult for health professionals to discuss end-of-life care when there is no support from the family or the patient themselves who is unable to contact the subject. It is essential that the team guides the family on the importance of this discussion and acts as an intermediary, interacting with the family member so that they can also express their wishes.
We added this reference:
[58] Santos, E. C. S. dos. Palliative Care and the Role of the Patient Through Advance Directives. Rev. Tec. Cient. CEJAM, 2024, 3, e202430027. https://doi.org/10.59229/2764-9806.RTCC.e202430027.
- Placement of Figure 1 and grey boxes at the top:
2a. I wondered whether Figure 1 should be presented a bit later, as it makes much more sense after explanations about the Go Wish Cards Game (GWCG). The last sentence in the introduction could remain the same, but without “(Figure 1)”. The figure could be added a bit later, perhaps with the addition of two or three sentences summarising its key points?
Response: Thank you for your insightful feedback regarding the placement of Figure 1. We carefully considered your suggestion and agree that positioning the figure later in the text significantly improves its clarity and relevance. As a result, we have moved Figure 1 to a later section, following the explanations about the GWCG, as you recommended. Additionally, we included sentences summarizing the key points of the figure to provide a clearer context and enhance its overall impact. We truly appreciate your thoughtful observations.
2b. Would the grey items on top look better in boxes? The way they are displayed, it looks like there is a specific order/sequence to them. From what I understood this is not the case and this is just a list (in no particular order)?
Response: Thank you for your thoughtful observation regarding the grey items at the top of Figure 1. We agree with your assessment, and you are absolutely correct—there is no specific order or sequence to these barriers. To address this, we have placed them in boxes, as you suggested, and removed the sequential arrows that previously gave the impression of a progression. This change ensures that the list is presented more clearly, reflecting that the barriers are not in any particular order. We truly appreciate your valuable input, which has helped improve the clarity of the figure.
- Page 3 of 12, line 104: it may be best to add a subject to the sentence, either repeating GWCG, or if wishing to avoid repetition, “The cards game consists of…”
Response: We have adjusted on page 3 - The game consists of 36 cards
- Page 7 of 12, line 276: Perhaps “family factors” instead of “familial factors”? The latter seems to indicate this is about diseases or disorders that run in families.
Response: Thank you for your suggestion. We have adjusted (page 7)
- Table 1, page 8 of 12: I like this table, but it is not clear if it is derived solely from the evidence, if the resolution strategies have been developed by the manuscript authors, or maybe a combination of both. It would be useful to clarify this, even if as a footnote.
Response: Thank you for your suggestion. The guidelines described in Table 1 are derived from the main scientific evidence found in this narrative review, in order to highlight important strategies for implementing the GWCG in clinical practice, despite its challenges and barriers. A footnote has been inserted in Table 1 (page 10).
- Overall: Are the contents from the cards copyrighted? A text box with the items would be very informative, although I understand this may not be possible.
Response: Thank you for your suggestion. We added in the text: “Go WishÒ was developed by Coda Alliance. In some countries, there are partners authorized to sell the game, including a guide for its use.” (page 3)
We sincerely appreciate all the suggestions and feedback, which were crucial in helping us improve the article